# Matching Standard and Secant Parallels in Cylindrical Projections

**Miljenko Lapaine**

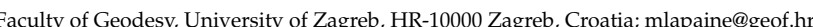

Faculty of Geodesy, University of Zagreb, HR-10000 Zagreb, Croatia; mlapaine@geof.hr

**Abstract:** Map projections are usually interpreted by mapping a sphere onto an auxiliary surface, and then the surface is developed into a plane. It is taken as a fact without proof that the parallels in which the auxiliary surface intersects the sphere are mapped without distortions. In a previous paper, based on a theoretical consideration and illustrated with several examples, the author concluded that explaining cylindrical projections as mapping onto a cylindrical surface is not a good approach, because it leads to misunderstanding important properties of projection. In this paper I prove that there are no equal-area, equidistant, or conformal cylindrical projections for which the standard parallel will coincide with secant parallel after folding the map into a cylinder.

**Keywords:** map projection; standard parallel; secant parallel; cylindrical projection

## 1. Introduction

Map projection is the mapping of a curved surface, such as a sphere or ellipsoid, into a plane. The changes that occur in such a mapping are called distortions. We distinguish distortions of distances, surfaces, and angles. If there are no distortions at every point and in all directions of a line, we say that it is a line with zero distortion or a standard line. If this is true for a parallel, we say it is the standard parallel. At first glance, these are well-known, generally accepted definitions, but they are often confused with secant lines or parallels [1].

Although the need to distinguish between secant and standard parallels has been written about several times [2–6], this issue needs to be clarified repeatedly and additionally because the authors are either unaware of or lack the courage to accept it (e.g., [7–13].

In a previous paper [14] it was shown that the images of standard parallels after bending the map into a cylindrical surface and secant parallels generally do not match. In this paper we prove that there are no equal-area, equidistant, or conformal cylindrical projections for which the standard parallel will coincide with secant parallel after folding the map into cylinder.

In another paper [15] the geographic parameterization of a sphere with a radius $R$ and the center located in the origin of the coordinate system was introduced. Furthermore, the authors defined map projection, local linear scale factor in general, and specifically $h$ and $k$, local linear scale factors along a meridian, and a parallel, respectively:

$$h = \frac{\sqrt{E}}{R}, \; k = \frac{\sqrt{G}}{R\cos\varphi},\tag{1}$$

where

$$E = \left(\frac{\partial x}{\partial \varphi}\right)^2 + \left(\frac{\partial y}{\partial \varphi}\right)^2, \; G = \left(\frac{\partial x}{\partial \lambda}\right)^2 + \left(\frac{\partial y}{\partial \lambda}\right)^2,\tag{2}$$

$$x = x(\varphi, \lambda), \; y = y(\varphi, \lambda),\tag{3}$$

the geographic coordinates are $\varphi \in \left[-\frac{\pi}{2}, \frac{\pi}{2}\right]$, $\lambda \in [-\pi, \pi]$, as usual, and $x$ and $y$ are coordinates of a point in a rectangular (mathematical, right oriented) coordinate system

in the plane of projection. For simplicity, and without a loss of generality, we will choose $R = 1$ in this paper. Instead of relation (1) we can write

$$h = \sqrt{E}, \; k = \frac{\sqrt{G}}{\cos \varphi}. \tag{4}$$

If in all points of a parallel $k = 1$ is valid, regardless of the value of $h$, then the parallel has zero distortion in all points *in the direction of the parallel*, i.e., it is an *equidistantly mapped parallel*.

If in all points of a parallel $h = k = 1$ is valid, then the parallel has zero distortion in all points and *all directions*, i.e., it is a *standard parallel*. Thus, every standard parallel is an equidistantly mapped parallel, but the reverse is generally not true.

## 2. Standard and Secant Parallels in Normal Aspect Cylindrical Projections

The normal aspect of cylindrical projection of a sphere of radius 1 is a mapping defined by formulas

$$x = n\lambda, \; y = y(\varphi), \tag{5}$$

where $\varphi \in \left[-\frac{\pi}{2}, \frac{\pi}{2}\right]$, $\lambda \in [-\pi, \pi]$, $n > 0$ is a constant, and function $y = y(\varphi)$ is continuous, differentiable, monotone increasing and odd. As with any map projection, $x$ and $y$ are the coordinates of a point in a rectangular (mathematical, right-oriented) coordinate system in a plane. Anyone can see that this is a mapping to the plane, not to the surface of a cylinder. For such a mapping we have coefficients of the first fundamental form

$$E = \left(\frac{dy}{d\varphi}\right)^2, G = n^2. \tag{6}$$

Local linear scale factors along a meridian and a parallel, respectively, are

$$h = h(\varphi) = \frac{dy}{d\varphi}, \; k = k(\varphi) = \frac{n}{\cos \varphi}. \tag{7}$$

For the normal aspect cylindrical projection along the *equidistantly mapped parallel* to which the latitude $\varphi_1$ corresponds, it must be valid

$$k(\varphi_1) = 1, \tag{8}$$

or

$$\frac{n}{\cos \varphi_1} = 1 \text{ i.e., } n = \cos \varphi_1. \tag{9}$$

For the normal aspect cylindrical projection along the *standard parallel* to which the latitude $\varphi_1$ corresponds, it must be valid (8) and

$$h(\varphi_1) = 1, \tag{10}$$

or considering (7)

$$\frac{dy}{d\varphi}(\varphi_1) = 1. \tag{11}$$

Let us suppose that the map is produced in the normal aspect cylindrical projection given by (5). Let us fold this map into the cylindrical surface and mark the radius of that cylinder with $r$. Let us place that cylinder so that it cuts the unit sphere in two parallels ($\varphi_2$ and $-\varphi_2$, see Figure 1). These are *secant parallels*.

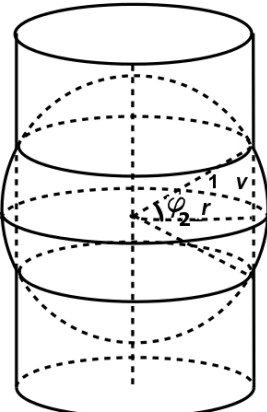

**Figure 1.** Map in a normal aspect cylindrical projection bent into a cylinder surface that intersects a sphere of radius 1, $r$ is the cylinder radius, $v$ is the height of secant parallel above the plane of the equator.

The circumference of the circle that is the base of this cylinder will be $2r\pi$, and at the same time it is equal to the length the image of the equator, or any parallel in the plane of projection, which is $2n\pi$ because for normal aspect cylindrical projections $x = n\lambda$. Hence, it immediately follows that $r = n$. Let us denote by $v$ the height above the equatorial plane in which the cylindrical surface intersects the sphere of radius 1.

Obviously, it is

$$\sin \varphi_2 = v, \ \cos \varphi_2 = n. \tag{12}$$

We see that the radius of the cylinder and the position of the intersecting parallels depend only on the parameter $n$ and will be the same for all cylindrical projections regardless of the function $y = y(\varphi)$.

Now, there is a question: if, in some normal aspect cylindrical projection, the parallel $\varphi_1$ is *mapped equidistantly*, will the image of this parallel coincide with the secant parallel after folding the map into the cylinder?

The distance between these two parallels on the map is $2y(\varphi_1)$. This distance remains unchanged after folding the map into a cylinder. So, the question is, is it worth for any normal aspect cylindrical projection

$$y(\varphi_1) = v = \sqrt{1 - n^2} \ ? \tag{13}$$

In a previous paper [14] it was shown on several examples that (19) is not valid in general. In the next section we will prove that relation (19) is not valid for any equal-area, equidistant or conformal normal aspect cylindrical projection.

### 3. Proof That Standard and Secant Parallels Do Not Match for Any Equal-Area, Equidistant or Conformal Cylindrical Projection

Lapaine (2022) showed that for some equal-area, equidistant and conformal cylindrical projections the standard and secant parallels do not match. In this section we will prove that this is also true for all equal-area, equidistant, or conformal cylindrical projections.

#### 3.1. Equal-Area Cylindrical Projections

The equations of any normal aspect cylindrical projection of a sphere of radius 1 are given by Formulas (5). The condition of equal-area mapping is

$$h(\varphi)k(\varphi) = \frac{n}{\cos \varphi}\frac{dy}{d\varphi} = 1, \tag{14}$$

from which we get

$$\frac{dy}{d\varphi} = \frac{\cos \varphi}{n}, \tag{15}$$

$$y = \frac{\sin \varphi}{n} + C, \tag{16}$$

where we denoted the additive constant by $C$. The oddness requirement for the function $y$ gives $C = 0$, so the equations of any equal-area cylindrical projection are:

$$x = n\lambda, \; y = \frac{\sin \varphi}{n}. \tag{17}$$

Let us suppose that $0 < n < 1$. From $y(\varphi_1) = \frac{\sin \varphi_1}{n}$ and (13) follows $\sqrt{1 - n^2} = n\sqrt{1 - n^2}$, which is possible only for $n = \pm 1$, and which is contrary to the assumption $0 < n < 1$. Thus, we have proved that there is no equal-area cylindrical projection of the form (17) for which the equidistantly mapped parallel will coincide with secant parallel after folding the map into cylinder. The same holds for the standard parallel, because a standard parallel is always equidistantly mapped one.

### 3.2. Equidistant Cylindrical Projections

The equations of any normal aspect cylindrical projection of a sphere of radius 1 are given by Formulas (5). The condition of equidistance along meridians is

$$h(\varphi) = \frac{dy}{d\varphi} = 1, \tag{18}$$

from which we get

$$y = \varphi + C, \tag{19}$$

where we denote the additive constant by $C$. The oddness requirement for the function $y$ gives $C = 0$, so the equations of any normal aspect equidistant cylindrical projection are:

$$x = n\lambda, \; y = \varphi. \tag{20}$$

Let us suppose that $0 < n < 1$. From (13) and $y(\varphi_1) = \varphi_1 = \sqrt{1 - n^2}$ follows $\varphi_1 = \sin \varphi_1$, which is possible only for $\varphi_1 = 0$, or $n = \pm 1$, and this is contrary to the assumption $0 < n < 1$. Thus, we have proved that there is no normal aspect equidistant cylindrical projection of the form (20) for which the equidistantly mapped parallel will coincide with secant parallel after folding the map into cylinder. The same holds for the standard parallel, because a standard parallel is always an equidistantly mapped one.

### 3.3. Conformal Cylindrical Projections

The equations of any normal aspect cylindrical projection of a sphere of radius 1 are given by Formulas (5). The condition of conformality reads

$$h(\varphi) = k(\varphi) = \frac{n}{\cos \varphi} = \frac{dy}{d\varphi}, \tag{21}$$

from which we get

$$y = \int \frac{n d\varphi}{\cos \varphi} + C = n \tan \mathrm{h}^{-1}(\sin \varphi) + C, \tag{22}$$

where we denote the additive constant by $C$. The oddness requirement for the function $y$ gives $C = 0$, so the equations of any normal aspect conformal cylindrical projection are:

$$x = n\lambda, \; y = n \tan \mathrm{h}^{-1}(\sin \varphi). \tag{23}$$

Let us suppose that $0 < n < 1$. From (13) and $y(\varphi_1) = n \tan \mathrm{h}^{-1}(\sin \varphi_1)$ it follows $n \tan \mathrm{h}^{-1}\sqrt{1 - n^2} = \sqrt{1 - n^2}$, or

$$\sqrt{1 - n^2} = \tan \mathrm{h} \frac{\sqrt{1 - n^2}}{n}. \tag{24}$$

The Equation (24) has only one solution, $n = 1$. This is not obvious and should be proved. After squaring the relation (24) we will get

$$1 - \tan h^2 \frac{\sqrt{1-n^2}}{n} = n^2. \tag{25}$$

From there we have

$$\frac{1}{\cos h^2 \frac{\sqrt{1-n^2}}{n}} = n^2 \tag{26}$$

and then

$$\cos h^2 \frac{\sqrt{1-n^2}}{n} = \frac{1}{n^2}. \tag{27}$$

It is known that it is

$$\cos hx = 1 + \frac{x^2}{2} + \frac{x^4}{24} + \cdots \tag{28}$$

and then

$$\cos h^2 x = 1 + x^2 + \cdots \tag{29}$$

Based on (27) and (29) we can write

$$\cos h^2 \frac{\sqrt{1-n^2}}{n} = 1 + \frac{1-n^2}{n^2} + \cdots = \frac{1}{n^2} + \cdots > \frac{1}{n^2} \tag{30}$$

This proves that Equation (24) has no solution for $0 < n < 1$. Thus, we have proved that there is no normal aspect conformal cylindrical projection of the form (23) for which the equidistantly mapped parallel will coincide with the secant parallel after folding the map into a cylinder. The same holds for the standard parallel, because a standard parallel is always an equidistantly mapped one.

## 4. Conclusions

Based on derivations in this paper, it can be concluded that the radius of the cylinder in which we bend the map made in a normal aspect cylindrical projection depends only on the parameter $n$ from the equation $x = n\lambda$. This means that all cylindrical projections with the same parameter $n$ have the same secant parallels located at a distance $v = \sqrt{1-n^2}$ above and below the equatorial plane. Since the various normal aspect cylindrical projections have differently spaced standard parallels, with remained unchanged distances when bending, it is clear that there will generally be no matching of the secant and standard parallels.

Conversely, suppose a cylinder is positioned so that it intersects a sphere in two parallels. Once we develop the surface of the cylinder into a plane, the spacing between the images of the intersecting parallels will not change. If by developing the cylinder surface the secant parallels became standard parallels, then all projections with the same secant parallels would have equally spaced standard parallels. That, of course, is not true.

We have proved that there is no equal-area, equidistant, or conformal cylindrical projection for which the standard and secant parallels coincide. Standard and secant parallels are often considered identical, but this paper shows that widely accepted facts about these parallels are wrong and need to be revised. This requires a critical approach to the established customs in teaching and researching map projections. It is always difficult to introduce changes when long-established custom has created a rut, but to prevent misunderstandings in the theory of map projections, we recommend avoiding the use of developable surfaces as intermediate surfaces, and thus secant parallels and projections.

**Funding:** This research received no external funding.

**Data Availability Statement:** Not applicable.

**Acknowledgments:** The author would like to thank Željka Tutek and Iva Kodrnja for their help in finding the solution of the Equation (24).

**Conflicts of Interest:** The author declares no conflict of interest.

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
