# Peer review of "Matching Standard and Secant Parallels in Cylindrical Projections"

_ijgi, doi:10.3390/ijgi12020063_

Round 1

Reviewer 1 Report

This is a high-quality paper. 

I suggest it be accepted in its current form.

Author Response

Thank you for the review. The paper was proofread by a native speaker.

Reviewer 2 Report

The paper aims to prove that there are no equal-area, equidistant, or conformal cylindrical projections for which the standard parallel coincide with secant parallels after folding the map to cylinder. The paper is written very shortly, making it a litter difficulty to understand the research purpose and the formula proofs presented by author. To my limited knowledge, standard parallel is the parallel without any distortion and for typical normal aspect cylindrical projections, the tangent or secant parallel is the standard parallel without distortion. So, I am confused about “coincide with”, please explain and you’d better to use a figure to illustrate it.

There are some detailed comments and minor mistakes listed as follows.

1.     Line 83: why the distance between these two parallels is 2y(φ1)? Is it meaning the distance between φ1 and φ2 or between φ1 and -φ1?

2.     Line 86-88: What is formula (13) meaning? And (19) is wrongly numbered?

3.     Line 133: equation (33) and (35) may be wrongly referenced.

4.     Maybe you can provide a link for the submitted literatures of [14].

Reviewer 3 Report

I think the misunderstanding lies in equation (13), which does not hold actually. The secant parallel has zero distortion in three different cylindrical projections.

Round 2

Reviewer 2 Report

I suggest it be accepted in its current form.

Reviewer 3 Report

After the second round review process, I understand the purpose of the paper and have changed the long-established ideas that standard parallel and secant parallel is the same thing.